# Salivary levels of Streptococcus mutans and Lactobacilli and other salivary indices in patients wearing clear aligners *versus* fixed orthodontic appliances: An observational study

**Stefano Mummolo[1]**, **Alessandro Nota** [2]* , **Francesca Albani[1]**, **Enrico Marchetti[1]**, **Roberto Gatto[1]**, **Giuseppe Marzo[1]**, **Vincenzo Quinzi[1]**, **Simona Tecco[2]**

1 Department of Life, Health and Environmental Sciences, University of L'Aquila, L'Aquila, Italy, 2 Dental School, Vita-Salute San Raffaele University and IRCCS San Raffaele Hospital, Milan, Italy

☯ These authors contributed equally to this work.
* dr.alessandro.nota@gmail.com

**Data Availability Statement:** All relevant data are within the paper and its Supporting Information files.

## Abstract

### Objective

This study aimed to investigate salivary levels of *Streptococcus mutans* (S. mutans) and Lactobacilli, and other salivary indices in subjects wearing clear aligners (CA) in comparison with multibrackets orthodontic appliances (MB).

### Materials and methods

A sample of 80 participants (46 males and 34 females) was included in the study: 40 subjects (aged 20.4±1.7 years) were treated with CA, and 40 (aged 21.3±1.7 years) were treated with MB. Plaque index (PI), salivary flow, buffering power of saliva, and salivary levels of S. mutans and Lactobacilli were evaluated prior to start of orthodontic treatment (t0), after 3 months (t1) and 6 months (t2).

### Results

CA patients maintained PI at level 0 over time, while MB participants experienced a statistically significant increasing trend of PI over time. In addition, at t2, 37.5% of MB participants (15 subjects over 40) showed risky salivary levels (CFU/ml>$10^5$) of S. mutans (odds ratio = 7.40; 95% C.I. = 1.94–28.25; chi-square = 10.32; p = 0.001) as well as Lactobacilli (odds ratio = 23.40; 95% C.I. = 2.91–188.36; chi-square = 15.31; p = 0.0001).

### Conclusions

Comparing all the data, subjects treated with CA achieved lower salivary microbial colonization after 6 months of treatment compared with MB. Different additional strategies for plaque

**Funding:** The author(s) received no specific funding for this work.

**Competing interests:** The authors have declared that no competing interests exist.

control and salivary microbial colonization must be triggered considering the type of orthodontic appliance.

## Introduction

The bacterial microflora, present in the oral cavity, contributes to the health of the host and prevents infections by potentially pathogenic exogenous microorganisms, thus providing resistance to the colonization of these parasitic species, and by regulating the inflammatory response towards the commensal bacteria that host in the buccal cavity.[1] Two literature reviews [2, 3] showed that there is moderate-to-high evidence that orthodontic appliances are able to significantly influence the concentration of oral microbiota, causing an alteration of the quantity of *Streptococcus mutans* (S. mutans) and Lactobacilli that can basically affect the process of tooth enamel demineralization [4], due to their acid production and tooth adhesive properties. This statement is confirmed both for removable [5] and fixed [6] orthodontic appliances.

From a clinical point of view, as a consequence of the changes in microbiota, enamel demineralization increases during orthodontic treatment with multibrackets appliances (MB) ranging in orthodontic patients from 2% to 97%. [7]

In general, it was also observed that removable appliances have less impact on the oral microbiota than fixed ones [5], but literature still lacks data on clear aligners (CA), an appliance that has become a highly demanded alternative to MB, due to its desirable characteristics providing aesthetic and comfortability [8]. Furthermore, considering previous observations for removable appliances [3, 5] there is a general clinical trend to prefer CA to maintain a more satisfactory level of oral hygiene during the orthodontic treatment, in respect to MB. A previous study compared the periodontal health status between subjects in orthodontic treatment with CA and MB, observing a better periodontal health in the group treated with CA [9]. On the basis of this background, the purpose of the present study was to investigate salivary levels of S. mutans, Lactobacilli and other salivary indices in patients wearing CA versus MB.

## Materials and methods

This is an observational prospective controlled study, aimed to investigate salivary levels of S. mutans, Lactobacilli and other salivary indices in patients wearing CA versus MB. The participants were selected from a population of young adult patients who were going to be treated for their malocclusion in a dental clinic in the geographical region of Abruzzo (Central Italy). A total sample of 80 subjects were selected, 40 participants were treated with CA (Invisalign, Align Technology, Santa Clara, CA, USA) and the other 40 participants (matched with the previous group for age and gender distribution) were treated with MB (Damon Q2, Ormco, Washington, DC, USA) (Table 1).

**Table 1.  Demographic information about the whole sample.**

|  | Clear aligners (CA group) | Multibrackets appliance (MB group) |
|---|---|---|
| **Gender** | 24 M and 16 F | 22 M and 18 F |
| **Numerosity** | 40 | 40 |
| **Age (mean ±sd)** | 20.4±1.7 years | 21.3±1.7 |

The orthodontic technique adopted for the treatment of each patient was chosen independently by the expert orthodontists prior to the beginning of this research project. The selection of participants was made on the basis of the following inclusion criteria: permanent dentition, adult age, complete dentition, and a malocclusion characterized by Angle class I, with low-middle level of crowding (the patients did not require any orthodontic teeth extraction). The following parameters were taken as exclusion criteria: presence of oral breathing, presence of active caries, chronic periodontitis, presence of prosthetic rehabilitations, presence of endodontically treated teeth, history of high grade gingivitis, poor oral hygiene, evaluated through initial Plaque index (PI) and Bleeding index (BI). Regarding dental caries as an exclusion criteria, they were assessed by clinical examination with specillum and mirror before the beginning of the study and during the study at each appointment. In particular, the orthodontic treatment was not initiated (brackets or first aligner positionment) if caries were found on the first clinical examination, in this case a conservative treatment of dental caries was performed before beginning the treatment. No caries were then observed in the whole sample during the study period.

A sample power analysis was carried out considering the results of a preliminary pilot study on the first 30 participants (cit.), recording a difference of 29% in the number of participants with a bacterial count for S. mutans higher than $10^5$ CFU/ml between the groups, concluding that a minimum sample size in order to achieve a power of 80% with an alpha of 0.05 should be 39 subjects for each group. All the participants were treated by two expert orthodontists, who have exclusively practiced the branch of orthodontics for more than 5 years. The enrollment of the participants was done from January 2016 to June 2017. The matching criteria were confirmed since no significant difference was detected in gender distribution and mean age between the participants of the two groups.

Data collection and follow-up of the subjects were performed in the following way. A proper informed consent form was signed by each subject during one of the preliminary appointments before beginning the actual treatment. To motivate the participants, a free oral hygiene session (with instructions) was offered. Furthermore, subsequent follow-up visits included in the present protocol were arranged free of charge. This clinical protocol was in accordance with the ethical standards reported in the Declaration of Helsinki of 1975, and the study was ethically approved by the Ethical Committee of the University of L'Aquila (Document DR206/2013).

As the home oral hygiene habits could be potential confounders, a few days before the beginning of the observational period a professional oral hygiene procedure was performed to each of the participants, and accurate home oral hygiene instructions were given to each subject to be employed at home. Then, on the day scheduled for the beginning of the orthodontic treatment (T0), a salivary sample was taken from each subject before the bonding procedure, and other salivary samples were also taken after periods of 3 months (T1) and 6 months (T2). All the salivary samples were taken by the same operator during the morning (9am-12am).

The salivary samples were analyzed by *CRT® prevention system* (Ivoclar Vivadent Clinical, Schaan, Liechtenstein). the *CRT® buffer* system was used to evaluate salivary flow and saliva buffering power, and the *CRT® bacteria* was employed for the bacterial count, as previously published [6, 10]. At each of the follow-up appointments participants were recommended to refrain from eating, drinking alcohol and brushing their teeth for at least one hour prior to the visit, as all these actions could alter the average salivary flow. During every scheduled appointment, first the PI was recorded (clinical recordings were made by the author S.M.). Then, the patient was asked to chew a stimulant paraffin tablet for 30 seconds, then the secreted saliva was collected. These samples were immediately subjected to evaluation of the buffering capacity by using the *CRT® buffer*, which was expressed, according to the manufacturer's

instructions, in five scores. Subsequently, the participant was invited to chew the paraffin tablet for another 5 minutes, collecting all the produced saliva in a scaled glass tube and then the flow rate was calculated (ml/min). Through this procedure, milliliters of saliva were collected and the salivary flow rate for minutes (ml/min) was calculated. Subsequently, a part of each sample was used for bacterial count through the *CRT® bacteria*. The saliva was placed in culture media (agar) using pipettes. A $NaHCO_3$ tablet was added to stimulate the growth of bacteria. The culture was then placed in an incubator at 35–37°C for 48 hours. S. mutans colonies appeared visible as small blue colonies with a diameter < 1mm on blue agar, while white color Lactobacilli colonies were detected on transparent agar. The presence of a bacterial count higher than $10^5$ CFU/ml of saliva indicates a high risk of developing caries (cut-off value for the high risk). Thus, in this study, subjects were dichotomized as S. mutans and Lactobacilli CFU > or < $10^5$ CFU/ml, which is considered the cut-off value for the high risk [11, 12].

### Data analysis

To avoid bias, the data were analyzed by operators who did not know the origin group of the collected data. The buffering power and the salivary flow were taken as nominal variables. In particular, for the salivary flow, the data were handled in 3 categories, as follows: level 1 = 0<ml<1.5; level 2 = 1.5<ml<2; and level 3 = ml>2. For the buffering power of saliva, the data were categorized in the following way: 1 = low; 2 = medium-low; 3 = medium; 4 = medium-high; 5 = high. These data were compared between the two groups using the chi-square test.

The data about the microbiota were examined considering the number of subjects (and percentage) with CFU/ml ≥ or < the cut-off value (*i.e.* $10^5$ CFU/ml). These percentages were compared over time at t0, t1 and t2, and differences between the two groups were investigated using the Binomial test.

The Plaque index (PI) was considered as a continuous variable and recorded as 0, 1, 2, and 3 values; for this variable, descriptive statistics showed mean and standard deviation; differences between the two groups were tested by *Student* t test.

For all the analyses the p value was set at 0.05.

### Results

All the participants completed the study, without any adverse event, and there were no missing data. Table 2 shows the distribution of values for the PI, the salivary flow and the buffering power of saliva at t0 in both groups.

For the salivary flow, a statistically significant difference was found between the two groups at t0, with the CA group showing a low level of salivary flow (0<ml<1.5) more frequently than the MB group (p = 0.01). Table 3 reports descriptive statistics (mean ± standard deviation) for the PI over time, at t0, t1 and t2, in the two groups. In the CA group, PI remained 0 over the whole follow-up period in all the subjects, while a progressive increase over time was observed in MB group, with a statistically significant increase over time, from t0 to t2, and significant differences with the other group at t2 and t3.

The buffering power of the saliva remained almost constant over the follow-up period in both the two groups (score 5).

Table 4 shows the average values of salivary flow. As seen, the two groups showed a statistically significant difference at t0 (p = 0.01), although the modal value resulted 1 in both the groups. At t1 and t2 no significant difference was observed between the two groups.

Table 5 shows the percentage of subjects with S. mutans and Lactobacilli CFU/ml >$10^5$ over time in both groups. In the CA group, the number of subjects with S.mutans CFU/ml

**Table 2. Data observed before the beginning of the study in the two groups.**

|  | Clear aligners (CA group) (modal value) N = 40 | Multibrackets appliance (MB group) (modal value) N = 40 | CA Group vs MB Group |
|---|---|---|---|
| **PI** | 0 | 0 | n.s. |
| **Salivary flow** | 1 (observed in 26 over 40 subjects; 65%) | 1 (observed in 21 over 40 subjects; 52.5%) | Chi-square: 13.61; p = 0.01 |
| **Buffering power** | 3 | 3 | n.s. |

For the plaque index, the scores are 0.1.2.3.

For salivary flow: low (1 = 0<ml<1.5); medium (2 = 1.5<ml<2); and high (3 = ml>2).

For the buffering power of saliva: 1 = low; 2 = medium-low; 3 = medium; 4 = medium-high; 5 = high.

$>10^5$ slightly increased at t2, without statistical relevance. In the MB group, it increased progressively over time, with a statistically significant difference from t0 to t1 and from t0 to t2. The differences between the two groups were statistically significant at t1 and t2. At t2 only 8% of CA subjects (8 subjects over 40) showed S.mutans CFU/ml $>10^5$, whereas 37.5% of MB subjects (15 subjects over 40) showed it. Thus, MB subjects manifested an odds ratio of 7.40 (95% C.I. = 1.94–28.25; chi-square = 10.323; p = 0.001) in respect to CA subjects, which suggests that MB subjects after 6 months of treatment are at a higher risk of developing caries, due to the salivary concentrations of S. mutans.

For the Lactobacilli CFU/ml $>10^5$ the differences between the two groups were also statistically significant at t1 and t2. Thus, the MB group manifested a 23.40 odds ratio(95% C.I. = 2.91–188.36; chi-square = 15.313; p = 0.0001) of showing risky values of Lactobacilli colonization after 6 months of treatment compared to the CA group.

## Discussion

This study aimed to investigate PI, the salivary levels of S. mutans and Lactobacilli, and other salivary indices in subjects wearing CA versus MB. In fact, scientific evidence showed that the early stage of caries is highly affected by S. mutans and by visible dental plaque on maxillary incisors whereas cavities are strongly related to lactobacilli [13].

From the present data, it results that PI did not show statistically significant changes over time in the CA group, while the MB group experienced a statistically significant increasing trend, with a clinically relevant PI at t2 (mean PI>1) as reported in Table 3, with statistically significant differences compared to the other group at t1 and t2. This result suggests that CA allow the maintenance of a better oral hygiene level, compared to MB. This observation agrees with previous data from literature, indicating that teenagers treated with CA show a significantly lower plaque index than those treated with MB, even at a follow up of 12 months [14]. In addition, a systematic review pointed out, with a level of moderate evidence, that periodontal health indexes are significantly improved with CA compared to MB [15].

**Table 3. Plaque index (mean and standard deviation) in the two groups at t0, t1 and t2, with intra-group and between groups differences.**

|  | t0 Initial | t1 After 3 months | t2 After 6 months | t0 versus t1 comparison | t0 versus t2 comparison | t1 versus t2 comparison |
|---|---|---|---|---|---|---|
| **Clear aligners (CA group)** | 0 | 0 | 0 | n.s. | n.s. | n.s. |
| **Multibrackets appliance (MB group)** | 0 | 0.7 ± 0.55 | 1.4 ± 0.5 | t = -7.85 p = 0.001 | t = -5.6 p = 0.001 | t = -17.8 p = 0.001 |
| **CA group versus MB group comparison** | n.s. | t = -7.85 p = 0.001 | t = -17.8 p = 0.001 |  |  |  |

n.s. = not significant

**Table 4. Salivary flow modal values (number and percentage of subjects with the modal values) at t0, t1 and t2 in the two groups, and significant differences.**

| | t0 Initial | t1 After 3 months | t2 After 6 months | t0 versus t1 comparison | t0 versus t2 comparison | t1 versus t2 comparison |
|---|---|---|---|---|---|---|
| **Clear aligners (CA group)** | 1 (26 over 40; 65%) | 1 (22 over 40; 55%) | 1 (22 over 40; 55%) | n.s. | n.s. | n.s. |
| **Multibrackets appliance (MB group)** | 1 (21 over 40; 52.5%) | 1 (24 over 40; 60%) | 1 (24 over 40; 60%) | n.s. | n.s. | n.s. |
| **CA group versus MB group Comparison** | Chi-square: 13.61; p = 0.01 | n.s. | n.s. | | | |

n.s. = not significant

Regarding the buffering power of saliva, described in Table 4, the present survey data show that it remained stable over time in both the two groups, as expected, because this is a physical property of saliva. It prevents the colonization of pathogenic microorganisms in the mouth and neutralizes acids produced by acidogenic bacteria, and enamel demineralization [10, 16]. The present data confirm previous reports in literature, which used the same methods to evaluate the buffering capacity of saliva in the course of removable orthodontic treatment [5] and with passive self-ligating brackets [6] and reported no change for this parameter over the follow up period. Thus, it could be stated that CA are unable to influence the buffering power of saliva.

Similarly, the salivary flow (Table 4) showed no difference between the two groups at t1 and t2. The great part of participants showed a normal value of salivary flow (<1.5 ml). This observation suggests that the type of orthodontic appliance is unable to influence the salivary flow. At the authors' knowledge no previous study was performed about the buffering power of saliva and salivary flow in subjects treated with CA.

For the microbial colonization, *CRT® bacteria* was used to determine the *S. mutans* and *Lactobacilli* count in saliva by means of selective culture media. The preparation of samples and incubation were carried out according to the step-by-step procedure as it was described in

**Table 5. Number (and percentage) of subjects with S. mutans and Lactobacilli CFU $>10^5$, at t0, t1 and t2, with intra-group and between-groups differences.**

| | | | | | | |
|---|---|---|---|---|---|---|
| **S.Mutans CFU $> 10^5$** | | | | | | |
| | t0 Initial | t1 After 3 months | t2 After 6 months | t0 versus t1 comparison | t0 versus t2 comparison | t1 versus t2 comparison |
| **Clear aligners (CA group)** | 0 over 40 (0%) | 0 over 40 (0%) | 3 over 40 (8%) | n.s. | n.s. | n.s. |
| **Multibrackets appliance (MB group)** | 0 over 40 (0%) | 8 over 40 (20%) | 15 over 40 (37.5%) | Chi-square = 8.88 p = 0.002 | Chi-square = 18.462 P = 0.001 | n.s. |
| **CA group versus MB group Comparison** | n.s. | Chi-square = 8.88 p = 0.002 | Chi-square = 10.32 p = 0.001 | | | |
| **Lactobacilli CFU $> 10^5$** | | | | | | |
| | t0 Initial | t1 After 3 months | t2 After 6 months | t0 versus t1 comparison | t0 versus t2 comparison | t1 versus t2 comparison |
| **Clear aligners (CA group)** | 0 over 40 (0%) | 0 over 40 (0%) | 1 over 40 (2.5%) | n.s. | n.s. | n.s. |
| **Multibrackets appliance (MB group)** | 0 over 40 (0%) | 8 over 40 (20%) | 15 over 40 (37.5%) | Chi-square = 8.88 p = 0.002 | Chi-square = 18.462 p = 0.001 | n.s. |
| **CA group versus MB group Comparison** | n.s. | Chi-square = 8.88 p = 0.002 | Chi-square = 15.313 p = 0.0001 | | | |

n.s. = not significant

its instruction brochure. This test only determines whether or not S.mutans are present in dental saliva in quantity that determines an increase of the caries risk.

The CRT package can be considered a comprehensive test, whose main benefits are to determine the caries risk status, to create the basis for target treatment and individualized check-up intervals for the long-term maintenance of oral health. This chair-side method is highly specific and sensitive for S. mutans and it is only limited by the 48 hours required for detection of S.mutans [17].

A different trend of bacterial colonization in the two groups was observed (Table 5). All the CA participants showed S. mutans and Lactobacilli colonies under the risky values (i.e. $10^5$ CFU/ml) at t0 and t1. While at t2, only 8% of the group (3 participants over 40) showed a risky value. While the MB group showed a progressive increase over time from t0 to t1 (p<0.01) and at t2, when the 37.5% of participants (15 participants over 40) showed risky values. As seen, the count of Lactobacilli colonies showed a trend similar to S. mutans (Table 5) in the two groups. These trends suggest that CA do not increase the bacterial growth, and consequently the caries risk, in about 90% percent of the treated subjects, even after six months of treatment. These results are in agreement with a recent study by Wang et al. (Wang et al. 2019) that performed a microbiological analysis of salivary bacteria in subjects in orthodontic treatment with CA compared with MB and untreated control group, observing a lower abundance of Firmicutes (the phylum of both S. Mutans and Lactobacilli) in the CA group compared with the MB group and the absence of differences with the control group. On the contrary, a recent prospective study, performed by quantitative polymerase chain reaction, found no differences in the identification of S. mutans and Lactobacillus acidophilus in the saliva of participants wearing thermoplastic retainers compared with subjects in orthodontic treatment with fixed appliances. But in that study no quantitative analysis was possible, as almost no Lactobacillus acidophilus was identified in the collected samples [18].

From a clinical point of view, comparing all the data, it can be stated that only about 8% of CA participants achieve risky values of microbiota colonies after 6 months of treatment, with a stable plaque control, differently from MB participants. The maintenance of a better macroscopically (plaque index) and microscopically (S. mutans and Lactobacilli CFU) oral hygiene levels in CA participants, compared with MB participants, should be related to the absence of retentive surfaces on the patient's teeth and the consequent facilitation of oral hygiene even comparable with an untreated subject [6]. While in about 40% of MB participants there is an increased risk of dental caries and demineralizations and so additional strategies for plaque control must be applied after the first six months of treatment (and for 20% just after 3 months), the use of CA seems to significantly limit this risk to less than 10% of the subjects. Additional strategies in subjects with risk of developing demineralization could be the use of disinfecting and mineralizing substances, to be used at all times when the patient has no social interactions, preserving the potential aesthetic of therapy [16]. An additional aid can be provided by the use of food, like some yoghurt that can positively influence the oral ecosystem [19].

Literature has not clarified yet whether there is a direct proportion between the count of bacterial colonies in the saliva and the appearance/increase of decalcifications on the teeth during orthodontic treatment. This correlation is actually derived from a clinical observation, according to which decalcifications near to orthodontic brackets can increase during orthodontic treatment [20], and from the laboratory observation that the number of bacterial colonies in the saliva is linked to a higher risk of enamel decalcification [11]. From one of the classical studies in literature dated 1989, S. mutans has been detected in the dental plaque near an orthodontic band both in areas interested by enamel dissolution and in other areas of the teeth not interested by enamel dissolution, without any relationship between the presence of S.

mutans and enamel dissolution [21]. Similarly, while it must be observed that the literature is poor about the topic, it seems that there is no direct relationship between salivary and dental plaque microbiome [22], both plaque amount and saliva microbiomes, analyzed in the present study, are associated with the development of dental caries [11, 23].

A considerable limitation of the present study, related with the absence of a proper randomization, is the presence of a selection bias related with the possible higher socioeconomic status of subjects treated with CA rather than MB considering that there was a reported association between oral hygiene level and socioeconomic status [24].

While exclusion criteria were carefully applied to exclude potential confounders [25], this study has some limitations because only subjects with Angle class I malocclusion with low/mild level of crowding were included, and consequently it wasn't possible to analyze the type of malocclusion as confounding factor and the present results are limited to young adult subjects with Angle class I malocclusion and low/mild level of crowding and further studies are needed to generalize them to other types of malocclusion. Similarly, no certain data were included about the compliance of the participants regarding home oral hygiene procedure, although proper instructions were given to all the participants. Furthermore, other limitations of the present study were the use of the *CRT® prevention system* (Ivoclar Vivadent Clinical, Schaan, Liechtenstein) as an in-office test rather than other laboratory tests with the related limitations, among them the use of paraffin wax to stimulate salivary flow, that may have slightly modified the results by detaching bacteria from surfaces.

## Conclusions

Comparing all the data, orthodontic treatment with CA appliances allow the maintenance of a better oral hygiene level, compared to MB. Only 8% of CA participants against approximately 40% of MB participants showed high concentrations of S. Mutans after 6 months of treatment requiring additional strategies for plaque control considering the type of orthodontic appliance.

## Supporting information

**S1 Data.**
(XLSX)

## Acknowledgments

The authors acknowledge Dr. Atanaz Darvizeh and Dr. Floriana Bosco for their precious help in the English proofreading of the manuscript.

## Author Contributions

**Conceptualization:** Stefano Mummolo, Alessandro Nota, Simona Tecco.

**Data curation:** Stefano Mummolo, Alessandro Nota, Francesca Albani, Simona Tecco.

**Formal analysis:** Stefano Mummolo, Alessandro Nota, Enrico Marchetti, Simona Tecco.

**Investigation:** Stefano Mummolo, Alessandro Nota, Francesca Albani, Simona Tecco.

**Methodology:** Stefano Mummolo, Alessandro Nota, Simona Tecco.

**Project administration:** Stefano Mummolo, Alessandro Nota, Roberto Gatto, Giuseppe Marzo, Vincenzo Quinzi, Simona Tecco.

**Supervision:** Enrico Marchetti, Roberto Gatto, Giuseppe Marzo, Vincenzo Quinzi.

**Writing – original draft:** Stefano Mummolo, Alessandro Nota, Francesca Albani, Enrico Marchetti, Simona Tecco.

**Writing – review & editing:** Stefano Mummolo, Alessandro Nota, Roberto Gatto, Giuseppe Marzo, Vincenzo Quinzi, Simona Tecco.

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
