## [Decision Letter · Decision Letter 0]

26 Sep 2019

PONE-D-19-21131

Salivary levels of Streptococcus mutans and Lactobacilli and other salivary indices in patients wearing clear aligners versus fixed orthodontic appliances: an observational study

PLOS ONE

Dear Dr. Nota,

Thank you for submitting your manuscript to PLOS ONE. After careful consideration, we feel that it has merit but does not fully meet PLOS ONE’s publication criteria as it currently stands. Therefore, we invite you to submit a revised version of the manuscript that addresses the points raised during the review process.

Please address the microbiology comments from reviewers and expand causality to further discuss your study's limitations and future studies that may address these issues. 

We would appreciate receiving your revised manuscript by Nov 10 2019 11:59PM. To enhance the reproducibility of your results, we recommend that if applicable you deposit your laboratory protocols in protocols.io, where a protocol can be assigned its own identifier (DOI) such that it can be cited independently in the future. For instructions see: http://journals.plos.org/plosone/s/submission-guidelines#loc-laboratory-protocols

We look forward to receiving your revised manuscript.

Kind regards,

Sompop Bencharit, DDS, MS, PhD, FACP

Academic Editor

PLOS ONE

Journal Requirements:

Reviewers' comments:

Reviewer's Responses to Questions

**Comments to the Author**

1. Is the manuscript technically sound, and do the data support the conclusions?

Reviewer #1: Yes

Reviewer #2: Yes

2. Has the statistical analysis been performed appropriately and rigorously? 

Reviewer #1: Yes

Reviewer #2: Yes

3. Have the authors made all data underlying the findings in their manuscript fully available?

Reviewer #1: Yes

Reviewer #2: Yes

4. Is the manuscript presented in an intelligible fashion and written in standard English?

Reviewer #1: No

Reviewer #2: Yes

5. Review Comments to the Author

Reviewer #1: Thank you for your submission

A few issues - firstly, the English is not easy to read, so i suggest you recruit a native English speaker knowledgeable in cariology or microbiology

the term 'patient' should be avoided and replaced with 'participant'

Streptococcus mutans in italics or S. mutans (space between S. and mutans) should be used - you have a variety of iterations of these

On page 4, you mention 'caries' incidence increases as well as white spot lesions - white spot lesions are carious lesions - please correct terminology - maybe refer to Innes et al JDR 2016 caries terminology paper

Did you attempt to categorise the amount of crowding apart from 'low-middle'? Crowding features such as rotations and imbrications can have an effect on oral hygiene effectiveness

Did you collect dietary data, as the type of appliance could influence dietary habits?

You do not discuss the limitations of the simplistic microbiological testing methodology you have used - also, is salivary microbiome representative of the plaque microbiome? this should be discussed

Are Lactobacilli involved in the early development of carious lesions?

In the cut off values for bacterial CFUs, you cite 4 papers, none of which investigated bacterial counts with carious lesion development - do you think these are appropriate references?

Is there any evidence that salivary buffering capacity tested as you have done is correlated with caries risk?

Did you examine the participants for carious lesions? if not - why not?

If you did, why isn't it reported?

Of the four references you use to justify the cut-oof values for CFUs, only one correlated Lactobacilli counts to carious lesions, and the majority of children had cavitated lesions - which i assume is not the case in your study- these are not sound references to use with early carious lesions

Reviewer #2: Is there any correlation between number of bacteria and decalcification lesions? if so, please discuss

6. PLOS authors have the option to publish the peer review history of their article (what does this mean?). If published, this will include your full peer review and any attached files.

Reviewer #1: No

Reviewer #2: Yes: Phimon Atsawasuwan

---

## [Author Response · Author response to Decision Letter 0]

12 Nov 2019

Reviewer #1: Thank you for your submission

A few issues - firstly, the English is not easy to read, so i suggest you recruit a native English speaker knowledgeable in cariology or microbiology

R: The manuscript was fully proofreaded by a native English speaker dentist

The term 'patient' should be avoided and replaced with 'participant'

R: thanks for the suggestion. It was changed all over the manuscript

Streptococcus mutans in italics or S. mutans (space between S. and mutans) should be used - you have a variety of iterations of these

R: thanks for the suggestion. It was changed all over the manuscript

On page 4, you mention 'caries' incidence increases as well as white spot lesions - white spot lesions are carious lesions - please correct terminology - maybe refer to Innes et al JDR 2016 caries terminology paper

R: thanks to the reviewer for this suggestion. We considered the terminology referring to Innes et al (2016) Innes NP, Frencken JE, Schwendicke F. Don't Know, Can't Do, Won't Change: Barriers to Moving Knowledge to Action in Managing the Carious Lesion. J Dent Res. 2016 May;95(5):485-6. doi: 10.1177/0022034516638512. PMID: 27099269)

Did you attempt to categorise the amount of crowding apart from 'low-middle'? Crowding features such as rotations and imbrications can have an effect on oral hygiene effectiveness

R: thanks to the reviewer for this suggestion. Unfortunately, in this study, we included only patients with "low-middle" degree of crowding, in order to maintain the homogeneity between the two groups. Surely this inclusion criterion represents a limit for the present study. The reviewer with this observation gives us an indication to future studies that also include patients with “moderate/ severe” crowding. This was also already reported among the limits of the study at the end of the discussion section.

Did you collect dietary data, as the type of appliance could influence dietary habits?

R: unfortunately we have not investigated patients' eating habits. The reviewer with this observation gives us an indication to future studies.

You do not discuss the limitations of the simplistic microbiological testing methodology you have used

R: thanks to the reviewer for the suggestion. The following sentence was added in the Discussion: “CRT® bacteria was used to determine the S. mutans and Lactobacilli count in saliva by means of selective culture media. The preparation of samples and incubation were carried out according to the step-by-step procedure as it was described in its instruction brochure. This test only determines whether or not S.mutans are present in saliva in quantity that determines an increase of the caries risk.

The CRT package can be considered a comprehensive test, whose main benefits are to determine the caries risk status, to create the basis for target treatment and individualized check-up intervals for the long-term maintenance of oral health. This chair-side method is highly specific and sensitive for S. mutans and its only limitation is that at least 48 hours are required for detection of S. mutans.

Sánchez-García S, Gutiérrez-Venegas G, Juárez-Cedillo T, Reyes-Morales H, Solórzano-Santos F, García-Peña C. A simplified caries risk test in stimulated saliva from elderly patients. Gerodontology. 2008;25:26–33. doi:10.1111/j.1741-2358.2007.00184.x.

Also, is salivary microbiome representative of the plaque microbiome? this should be discussed

R: The literature is poor on this topic and an old study showed that there isn’t a direct correlation between the plaque amount and the saliva microbiome (Jalil RA. Correlating Streptococcus mutans counts in saliva with plaque amount, gingival inflammation and caries experience in school children. Singapore Dent J. 1995 Jul;20(1):16-20.).

Anyway both plaque and saliva microbiomes are associated with the development of dental caries. In the present study the evaluation of the PI, and the saliva microbioma, gives to the literature a new step in this field.

This point was added to the discussion following the suggestion of the reviewer.

Are Lactobacilli involved in the early development of carious lesions?

R: The evidence remains weak or absent about the association between initial dental caries and Lactobacilli. (Neves BG, Stipp RN, Bezerra DDS, Guedes SFF, Rodrigues LKA.Quantitative analysis of biofilm bacteria according to different stages of early childhood caries.Arch Oral Biol. 2018 Dec;96:155-161. doi: 10.1016/j.archoralbio.2018.09.007.

Periodontol 2000. 2016 Feb;70(1):128-41. doi: 10.1111/prd.12100. Salivary biomarkers for dental caries. Gao X, Jiang S, Koh D, Hsu CY)

On the other hand it seems that the early stage of caries is highly affected by mutans streptococci and visible dental plaque on maxillary incisors whereas cavities are strongly related to lactobacilli that are then involved in the lesions development (Parisotto TM, Steiner-Oliveira C, Duque C, Peres RC, Rodrigues LK, Nobre-dos-Santos M. Relationship among microbiological composition and presence of dental plaque, sugar exposure, social factors and different stages of early childhood caries.Arch Oral Biol. 2010 May;55(5):365-73.) 

This was also added to the discussion in order to make more clear for the reader the involvement of each considered parameter.

In the cut off values for bacterial CFUs, you cite 4 papers, none of which investigated bacterial counts with carious lesion development - do you think these are appropriate references?

R: 3 out of 4 cited papers are previously published studies that uses the same microbiological technique and cutoff values for caries risk evaluation. Anyway, according to what suggested by the reviewer we removed these 3 citations and added another citation that observed the CFU cutoff value applied in the present study.

(Gábris K, Nagy G, Madléna M, Dénes Z, Márton S, Keszthelyi G, Bánóczy J.Associations between microbiological and salivary caries activity tests and caries experience in Hungarian adolescents. Caries Res. 1999 May-Jun;33(3):191-5.)

Is there any evidence that salivary buffering capacity tested as you have done is correlated with caries risk?

R: Salivary buffering capacity is correlated with caries risk as reported by different articles as Fernando S, Tadakamadla SK, Bakr M, Scuffham PA, Johnson NW. Indicators of Risk for Dental Caries in Children: A Holistic Approach. JDR Clin Trans Res. 2019 Oct;4(4):333-341. that used a similar caries risk analysis kit and was reported in the manuscript methods section.

The reference was also added to the manuscript.

Did you examine the participants for carious lesions? if not - why not?

If you did, why isn't it reported?

R: All subjects were examined for carious lesions and were free from caries before being included in the study and checked during the therapy.

We thank the reviewer for reminding us to include this as an inclusion criteria in the manuscript, it wasn’t previously reported because for every orthodontist this is a criteria for starting an orthodontic treatment and is considered a routine check.

Of the four references you use to justify the cut-off values for CFUs, only one correlated Lactobacilli counts to carious lesions, and the majority of children had cavitated lesions - which i assume is not the case in your study- these are not sound references to use with early carious lesions

R: As previosly stated, 3 out of 4 cited papers are previously published studies that uses the same microbiological technique and cutoff values for caries risk evaluation. Anyway, according to what suggested by the reviewer we removed these 3 citations and added another citation that observed the CFU cutoff value applied in the present study. (Gábris K, Nagy G, Madléna M, Dénes Z, Márton S, Keszthelyi G, Bánóczy J.Associations between microbiological and salivary caries activity tests and caries experience in Hungarian adolescents. Caries Res. 1999 May-Jun;33(3):191-5.)

Reviewer #2:

This manuscript described to investigate salivary levels of Streptococcus mutans (S.mutans) and Lactobacilli, and other salivary indices in patients wearing clear aligners (CA) versus multibrackets orthodontic appliances (MB).The authors determined Plaque index (PI), salivary flow, buffering power of saliva, and

salivary levels of S.Mutans and Lactobacilli were evaluated before the beginning of the orthodontic treatment (t0), and after 3 months (t1) and 6 months (t2) from the beginning. The authors recruited 80 subjects with equal numbers of Mb and CA treatments. 

Overall this study is an interesting area of investigation; however, there were some limitations and questions that the author should clarify for the comprehensive understanding of the manuscript. 

1. Please provide the time of saliva collection since it affects the composition and flow of saliva. 

R: Saliva was collected during morning (9-12am). This data was added in the Material and Methods section.

2. Please discuss the association between number of bacteria and incidence of decalcification in the discussion.

R: Literature has not clarified yet, whether there is a direct proportion between the count of bacterial colonies in the saliva and the appearance/increase of decalcifications on the teeth during an orthodontic treatment. This correlation is actually derived from a clinical observation according to which decalcifications near to orthodontic brackets can increase during an orthodontic treatment (Morrier JJ. White spot lesions and orthodontic treatment. Prevention and treatment. Orthod Fr. 2014 Sep;85(3):235-44. doi: 10.1051/orthodfr/2014016. Epub 2014 Aug 28), and from the laboratory observation that the number of bacterial colonies correlates in the saliva is linked to a higher risk to develop enamel decalcification (Messer LB. Assessing caries risk in children. Aust Dent J. 2000;45:10–6.). While from one of the classical study in literature dated 1989, S.mutans has been detected in the dental plaque near an orthodontic band both in areas interested by enamel dissolution, than in other areas of the teeth not interested by enamel dissolution, without any relationship between the presence of S.mutans and enamel dissolution. (Boyar RM1, Thylstrup A, Holmen L, Bowden GH. The microflora associated with the development of initial enamel decalcification below orthodontic bands in vivo in children living in a fluoridated-water area.J Dent Res. 1989 Dec;68(12):1734-8.) 

3. Normally the orthodontic treatment will take about 12 months, would the author follow up after 6 months. If so, would the author observe any difference or trend of bacteria level?

R: Patients were followed for orthodontic treatment beyond the period of this study. However, no salivary sample measurements were performed. Unfortunately, therefore, no data are available on the saliva microbiota beyond the period included in the follow-up of this study. Future research protocols could include this monitoring beyond the time of 6 months. 

4. Would the author explain the randomization process of subjects?

R: Unfortunately no randomization was performed. The type of orthodontic treatment had been chosen by the clinician in agreement with the patient, but the subjects were included in the samples matching age and gender distributions as already reported in the materials and methods section.

Thank you for the opportunity to review this manuscript.

---

## [Decision Letter · Decision Letter 1]

12 Dec 2019

PONE-D-19-21131R1

Salivary levels of Streptococcus mutans and Lactobacilli and other salivary indices in patients wearing clear aligners versus fixed orthodontic appliances: an observational study

PLOS ONE

Dear Dr. Nota,

Thank you for submitting your manuscript to PLOS ONE. After careful consideration, we feel that it has merit but does not fully meet PLOS ONE’s publication criteria as it currently stands. Therefore, we invite you to submit a revised version of the manuscript that addresses the points raised during the review process.

Three main issues need to be addressed:

1) Please consider more recent publications (Reviewer 2),

2) Address concerns on study limitations and explanation on the limitations,

3) Also since there are only 2 bacterial species mentioned, please explain how the results would be applicable to the overall microbiome (Reviewer 2).

4) The writing and organization of the manuscript need to be improved.

We would appreciate receiving your revised manuscript by Jan 26 2020 11:59PM. To enhance the reproducibility of your results, we recommend that if applicable you deposit your laboratory protocols in protocols.io, where a protocol can be assigned its own identifier (DOI) such that it can be cited independently in the future. For instructions see: http://journals.plos.org/plosone/s/submission-guidelines#loc-laboratory-protocols

We look forward to receiving your revised manuscript.

Kind regards,

Sompop Bencharit, DDS, MS, PhD, FACP

Academic Editor

PLOS ONE

Reviewers' comments:

Reviewer's Responses to Questions

**Comments to the Author**

1. If the authors have adequately addressed your comments raised in a previous round of review and you feel that this manuscript is now acceptable for publication, you may indicate that here to bypass the “Comments to the Author” section, enter your conflict of interest statement in the “Confidential to Editor” section, and submit your "Accept" recommendation.

Reviewer #1: (No Response)

Reviewer #2: All comments have been addressed

2. Is the manuscript technically sound, and do the data support the conclusions?

Reviewer #1: Partly

Reviewer #2: Partly

3. Has the statistical analysis been performed appropriately and rigorously? 

Reviewer #1: Yes

Reviewer #2: Yes

4. Have the authors made all data underlying the findings in their manuscript fully available?

Reviewer #1: Yes

Reviewer #2: No

5. Is the manuscript presented in an intelligible fashion and written in standard English?

Reviewer #1: No

Reviewer #2: Yes

6. Review Comments to the Author

Reviewer #1: Thank you for your responses to our comments and questions. There are still outstanding issues with the paper

The written English is still poor

The examination for caries was raised - and you have included a statement that active caries was an exclusion criteria - how was this determined?

you still have not included caries data in the paper - if you are relating bacterial counts to carious lesion development, you should present the caries data and correlate it to the bacterial counts

Your comment regarding the validity of the CRT system may be challenged by microbiologists

Regarding the Table s- i believe that several could be combined (such as Tables 6 and 7), or entered into the text (such as Table 4)

Your conclusions state that the CA are at lower risk of caries development - you cannot conclude this as you have not provided caries data - all you can conclude is about what you have measured - two bacterial counts and saliva characteristics

Reviewer #2: A recent article published in American Journal of Orthodontics and Dentofacial Orthopedics demonstrated the oral microbiome comparison between fixed, clear aligner and control groups and showed a different result from this current study. Please refer to the AJODO study and discuss about the different result so the manuscript will provide more information.

The selection bias could affect the result and conclusion. There wa a reported relationship between oral hygiene status and socioeconomic status and attitude toward types of appliances. Please state the limitation of your subject recruitment in the discussion since there is no random sampling in the process.

The conclusion should be rewritten to be specific to the species studied in the manuscript.

7. PLOS authors have the option to publish the peer review history of their article (what does this mean?). If published, this will include your full peer review and any attached files.

Reviewer #1: No

Reviewer #2: No

---

## [Author Response · Author response to Decision Letter 1]

16 Jan 2020

Editor

Three main issues need to be addressed:

1) Please consider more recent publications (Reviewer 2),

We were happy to add the recent publication that was indicated by the reviewer 2 and was previously not included because it was published in November 2019 after the submission of this manuscript.

2) Address concerns on study limitations and explanation on the limitations,

Following the reviewers suggestions other study limitations principally involved with the absence of randomization and the use of the CRT system. Another limitation was the fact that only two bacterial species are mentioned

3) Also since there are only 2 bacterial species mentioned, please explain how the results would be applicable to the overall microbiome (Reviewer 2).

The conclusions were limited to the 2 considered bacterial species that are involved in caries development, while information about the overall microbiome are not obtainable from the results of the present study.

4) The writing and organization of the manuscript need to be improved.

In the present new version of the manuscript, we tried to better organize the entire manuscript

Thanks to the Editor for this suggestion. In this revised version of the manuscript we tried to improve the writing and the organization of the manuscript, mostly for the Discussion section, where the order of the paragraphs was changed. beginning the discussion with the data about the PI, and then about the other properties of saliva (the salivary flow and the buffering capacity) and finally discussing the data about the microbial colonization. Then limitations and conclusions were stated.

Reviewer #1: Thank you for your responses to our comments and questions. There are still outstanding issues with the paper

The written English is still poor

R: English was carefully revised.

The examination for caries was raised - and you have included a statement that active caries was an exclusion criteria - how was this determined?

R: “Regarding dental caries as an exclusion criteria, they were assessed by clinical examination with specillum and mirror before the beginning of the study, and during the study period at each appointment. In particular, the orthodontic treatment was not started (brackets or first aligner positionment) if caries were found at the first clinical examination, in that case, a conservative treatment of dental caries was performed before beginning the treatment. Consequently, no active caries were observed in the whole sample during the study period.”. This sentence regarding the caries management was added in the materials and methods section.

You still have not included caries data in the paper - if you are relating bacterial counts to carious lesion development, you should present the caries data and correlate it to the bacterial counts

R: As reported in the previous answer and now reported in the study, no caries were observed during the study period, for this reason no further caries data were reported. The relationship between carious lesion development and bacterial count come from the previous literature, in the present study, a 6 months follow-up in subjects with medium to very good oral hygiene (as related to exclusion criteria) did not show the development of carious lesions that anyway wasn’t one of the objectives of the study.

Your comment regarding the validity of the CRT system may be challenged by microbiologists

R: Thanks to the reviewer for this comment. The use of the CRT system was explicitely reported as a limitation of the study according to the suggestion of the reviewer. In particular, it was used because it is a method that every dentist can adopt in his/her daily clinical routine, to monitor his/her patients during orthodontic treatment in a very simple way, differently from microbiological tests that are more difficult to be used in the daily routine

Regarding the Table s- i believe that several could be combined (such as Tables 6 and 7), or entered into the text (such as Table 4)

R: Thanks to the reviewer for this suggestion. In the present version of the paper, table 4 was deleted, as its data were entered into the text, while tables 6 and 7 were combined

Your conclusions state that the CA are at lower risk of caries development - you cannot conclude this as you have not provided caries data - all you can conclude is about what you have measured - two bacterial counts and saliva characteristics

R: The conclusions were restructured deleting the sentence about dental caries and reporting only what observed about oral hygiene and bacterial count. 

Reviewer #2: 

A recent article published in American Journal of Orthodontics and Dentofacial Orthopedics demonstrated the oral microbiome comparison between fixed, clear aligner and control groups and showed a different result from this current study. Please refer to the AJODO study and discuss about the different result so the manuscript will provide more information.

R: We really thank the reviewer for showing us the publication of a very recent paper (November 2019) about oral microbiome in clear aligners treatment (Alterations of the Oral Microbiome in Patients Treated with the Invisalign System or with Fixed Appliances.” American journal of orthodontics and dentofacial orthopedics. 156.5 (2019): 633–640.). After a careful evaluation of the article that analyzed a much wider microbiome spectrum, they state that “changes in the microbiome associated with Invisalign treatment showed different effects on oral health, with some beneficial and some harmful” and in particular they observed some negative results about periodontal harmful bacteria. Regarding caries involved bacteria, perfectly in agreement with this study, they observed that “For the Invisalign group, the abundance of Firmicutes was less than the fixed appliance group (I vs F) with no difference than the control group (I vs C)” and Firmicutes represents the Phylum of both S. Mutans and Lactobacilli analyzed in this study, related with caries risk rather than periodontal aspects. Furthermore, the sample size of that study is small (7 subjects with aligners and 14 with fixed appliances) and it is probably underpowered, in fact, no sample power analysis was reported, thus, the trustworthiness of that study (especially when they state an absence of significant differences between the microbiota of the two groups) is limited, this could be the reason of its contrast with the most complete systematic review on this topic that showed a better periodontal health in subjects in orthodontic treatment with clear aligners. 

For this reason, we discussed the results of that recent study regarding caries related bacteria (i.e. Firmicutes - S. Mutans and Lactobacilli) that are the outcome of our study in in the proper section.

The selection bias could affect the result and conclusion. There was a reported relationship between oral hygiene status and socioeconomic status and attitude toward types of appliances. Please state the limitation of your subject recruitment in the discussion since there is no random sampling in the process.

R: Thank you for the interesting observation, it was added to the limitations of the study.

The conclusion should be rewritten to be specific to the species studied in the manuscript. 

R: The conclusions were rewritten in order to be specific to the outcomes of the present study.

---

## [Editor Report · Decision Letter 2]

24 Jan 2020

Salivary levels of Streptococcus mutans and Lactobacilli and other salivary indices in patients wearing clear aligners versus fixed orthodontic appliances: an observational study

PONE-D-19-21131R2

Dear Dr. Nota,

We are pleased to inform you that your manuscript has been judged scientifically suitable for publication and will be formally accepted for publication once it complies with all outstanding technical requirements.

With kind regards,

Sompop Bencharit, DDS, MS, PhD, FACP

Academic Editor

PLOS ONE

Additional Editor Comments (optional):

Thank you for your responses to the reviewers. All comments have been addressed.
---

## [Editor Report · Acceptance letter]

13 Apr 2020

PONE-D-19-21131R2 

Salivary levels of Streptococcus mutans and Lactobacilli and other salivary indices in patients wearing clear aligners *versus* fixed orthodontic appliances: an observational study. 

Dear Dr. Nota:

I am pleased to inform you that your manuscript has been deemed suitable for publication in PLOS ONE. Congratulations! Your manuscript is now with our production department. 

With kind regards,

on behalf of

Dr. Sompop Bencharit 

Academic Editor

PLOS ONE